# Unsupervised Representation Learning by Predicting Image Rotations

**Spyros Gidaris, Praveer Singh, Nikos Komodakis**
University Paris-Est, LIGM
Ecole des Ponts ParisTech
`{spyros.gidaris,praveer.singh,nikos.komodakis}@enpc.fr`

## Abstract

Over the last years, deep convolutional neural networks (ConvNets) have transformed the field of computer vision thanks to their unparalleled capacity to learn high level semantic image features. However, in order to successfully learn those features, they usually require massive amounts of manually labeled data, which is both expensive and impractical to scale. Therefore, unsupervised semantic feature learning, i.e., learning without requiring manual annotation effort, is of crucial importance in order to successfully harvest the vast amount of visual data that are available today. In our work we propose to learn image features by training ConvNets to recognize the 2d rotation that is applied to the image that it gets as input. We demonstrate both qualitatively and quantitatively that this apparently simple task actually provides a very powerful supervisory signal for semantic feature learning. We exhaustively evaluate our method in various unsupervised feature learning benchmarks and we exhibit in all of them state-of-the-art performance. Specifically, our results on those benchmarks demonstrate dramatic improvements w.r.t. prior state-of-the-art approaches in unsupervised representation learning and thus significantly close the gap with supervised feature learning. For instance, in PASCAL VOC 2007 detection task our unsupervised pre-trained AlexNet model achieves the state-of-the-art (among unsupervised methods) mAP of $54.4\%$ that is only 2.4 points lower from the supervised case. We get similarly striking results when we transfer our unsupervised learned features on various other tasks, such as ImageNet classification, PASCAL classification, PASCAL segmentation, and CIFAR-10 classification. The code and models of our paper will be published on: `https://github.com/gidariss/FeatureLearningRotNet`.

## 1 Introduction

In recent years, the widespread adoption of deep convolutional neural networks (LeCun et al., 1998) (ConvNets) in computer vision, has lead to a tremendous progress in the field. Specifically, by training ConvNets on the object recognition (Russakovsky et al., 2015) or the scene classification (Zhou et al., 2014) tasks with a massive amount of manually labeled data, they manage to learn powerful visual representations suitable for image understanding tasks. For instance, the image features learned by ConvNets in this supervised manner have achieved excellent results when they are transferred to other vision tasks, such as object detection (Girshick, 2015), semantic segmentation (Long et al., 2015), or image captioning (Karpathy & Fei-Fei, 2015). However, supervised feature learning has the main limitation of requiring intensive manual labeling effort, which is both expensive and infeasible to scale on the vast amount of visual data that are available today.

Due to that, there is lately an increased interest to learn high level ConvNet based representations in an unsupervised manner that avoids manual annotation of visual data. Among them, a prominent paradigm is the so-called *self-supervised learning* that defines an annotation free pretext task, using only the visual information present on the images or videos, in order to provide a surrogate supervision signal for feature learning. For example, in order to learn features, Zhang et al. (2016a) and Larsson et al. (2016) train ConvNets to colorize gray scale images, Doersch et al. (2015) and Noroozi & Favaro (2016) predict the relative position of image patches, and Agrawal et al. (2015) predict the egomotion (i.e., self-motion) of a moving vehicle between two consecutive frames. The

rationale behind such self-supervised tasks is that solving them will force the ConvNet to learn semantic image features that can be useful for other vision tasks. In fact, image representations learned with the above self-supervised tasks, although they have not managed to match the performance of supervised-learned representations, they have proved to be good alternatives for transferring on other vision tasks, such as object recognition, object detection, and semantic segmentation (Zhang et al., 2016a; Larsson et al., 2016; Zhang et al., 2016b; Larsson et al., 2017; Doersch et al., 2015; Noroozi & Favaro, 2016; Noroozi et al., 2017; Pathak et al., 2016a; Doersch & Zisserman, 2017). Other successful cases of unsupervised feature learning are clustering based methods (Dosovitskiy et al., 2014; Liao et al., 2016; Yang et al., 2016), reconstruction based methods (Bengio et al., 2007; Huang et al., 2007; Masci et al., 2011), and methods that involve learning generative probabilistic models Goodfellow et al. (2014); Donahue et al. (2016); Radford et al. (2015).

Our work follows the self-supervised paradigm and proposes to learn image representations by training ConvNets to recognize the geometric transformation that is applied to the image that it gets as input. More specifically, we first define a small set of discrete geometric transformations, then each of those geometric transformations are applied to each image on the dataset and the produced transformed images are fed to the ConvNet model that is trained to recognize the transformation of each image. In this formulation, it is the set of geometric transformations that actually defines the classification pretext task that the ConvNet model has to learn. Therefore, in order to achieve unsupervised semantic feature learning, it is of crucial importance to properly choose those geometric transformations (we further discuss this aspect of our methodology in section 2.2). What we propose is to define the geometric transformations as the image rotations by 0, 90, 180, and 270 degrees. Thus, the ConvNet model is trained on the 4-way image classification task of recognizing one of the four image rotations (see Figure 2). We argue that in order a ConvNet model to be able recognize the rotation transformation that was applied to an image it will require to understand the concept of the objects depicted in the image (see Figure 1), such as their location in the image, their type, and their pose. Throughout the paper we support that argument both qualitatively and quantitatively. Furthermore we demonstrate on the experimental section of the paper that despite the simplicity of our self-supervised approach, the task of predicting rotation transformations provides a powerful surrogate supervision signal for feature learning and leads to dramatic improvements on the relevant benchmarks.

Note that our self-supervised task is different from the work of Dosovitskiy et al. (2014) and Agrawal et al. (2015) that also involves geometric transformations. Dosovitskiy et al. (2014) train a ConvNet model to yield representations that are discriminative between images and at the same time invariant on geometric and chromatic transformations. In contrast, we train a ConvNet model to recognize the geometric transformation applied to an image. It is also fundamentally different from the egomotion method of Agrawal et al. (2015), which employs a ConvNet model with siamese like architecture that takes as input two consecutive video frames and is trained to predict (through regression) their camera transformation. Instead, in our approach, the ConvNet takes as input a single image to which we have applied a random geometric transformation (i.e., rotation) and is trained to recognize (through classification) this geometric transformation without having access to the initial image.

Our contributions are:

- We propose a new self-supervised task that is very simple and at the same time, as we demonstrate throughout the paper, offers a powerful supervisory signal for semantic feature learning.

- We exhaustively evaluate our self-supervised method under various settings (e.g. semi-supervised or transfer learning settings) and in various vision tasks (i.e., CIFAR-10, ImageNet, Places, and PASCAL classification, detection, or segmentation tasks).

- In all of them, our novel self-supervised formulation demonstrates state-of-the-art results with dramatic improvements w.r.t. prior unsupervised approaches.

- As a consequence we show that for several important vision tasks, our self-supervised learning approach significantly narrows the gap between unsupervised and supervised feature learning.

In the following sections, we describe our self-supervised methodology in §2, we provide experimental results in §3, and finally we conclude in §4.

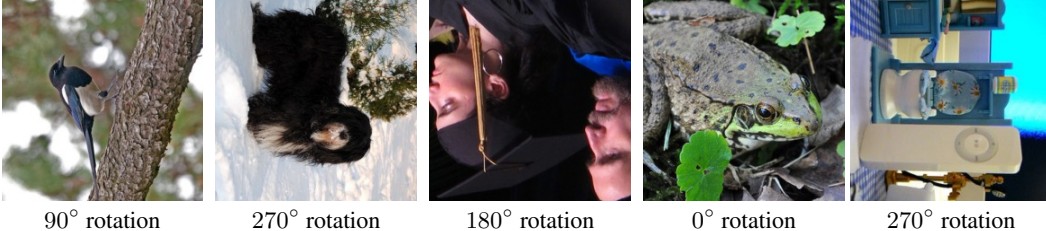

| 90° rotation | 270° rotation | 180° rotation | 0° rotation | 270° rotation |

Figure 1: Images rotated by random multiples of 90 degrees (e.g., 0, 90, 180, or 270 degrees). The core intuition of our self-supervised feature learning approach is that if someone is not aware of the concepts of the objects depicted in the images, he cannot recognize the rotation that was applied to them.

## 2 METHODOLOGY

### 2.1 OVERVIEW

The goal of our work is to learn ConvNet based semantic features in an unsupervised manner. To achieve that goal we propose to train a ConvNet model $F(.)$ to estimate the geometric transformation applied to an image that is given to it as input. Specifically, we define a set of $K$ discrete geometric transformations $G = \{g(.|y)\}_{y=1}^{K}$, where $g(.|y)$ is the operator that applies to image $X$ the geometric transformation with label $y$ that yields the transformed image $X^y = g(X|y)$. The ConvNet model $F(.)$ gets as input an image $X^{y^*}$ (where the label $y^*$ is unknown to model $F(.)$) and yields as output a probability distribution over all possible geometric transformations:

$$F(X^{y*}|\theta) = \{F^y(X^{y*}|\theta)\}_{y=1}^{K},\tag{1}$$

where $F^y(X^{y*}|\theta)$ is the predicted probability for the geometric transformation with label $y$ and $\theta$ are the learnable parameters of model $F(.)$.

Therefore, given a set of $N$ training images $D = \{X_i\}_{i=0}^{N}$, the self-supervised training objective that the ConvNet model must learn to solve is:

$$\min_{\theta} \frac{1}{N} \sum_{i=1}^{N} loss(X_i, \theta),\tag{2}$$

where the loss function $loss(.)$ is defined as:

$$loss(X_i, \theta) = -\frac{1}{K} \sum_{y=1}^{K} log(F^y(g(X_i|y)|\theta)).\tag{3}$$

In the following subsection we describe the type of geometric transformations that we propose in our work.

### 2.2 CHOOSING GEOMETRIC TRANSFORMATIONS: IMAGE ROTATIONS

In the above formulation, the geometric transformations $G$ must define a classification task that should force the ConvNet model to learn semantic features useful for visual perception tasks (e.g., object detection or image classification). In our work we propose to define the set of geometric transformations $G$ as all the image rotations by multiples of 90 degrees, i.e., 2d image rotations by 0, 90, 180, and 270 degrees (see Figure 2). More formally, if $Rot(X, \phi)$ is an operator that rotates image $X$ by $\phi$ degrees, then our set of geometric transformations consists of the $K = 4$ image rotations $G = \{g(X|y)\}_{y=1}^{4}$, where $g(X|y) = Rot(X, (y-1)90)$.

**Forcing the learning of semantic features:** The core intuition behind using these image rotations as the set of geometric transformations relates to the simple fact that it is essentially impossible for a ConvNet model to effectively perform the above rotation recognition task unless it has first learnt to recognize and detect classes of objects as well as their semantic parts in images. More specifically,

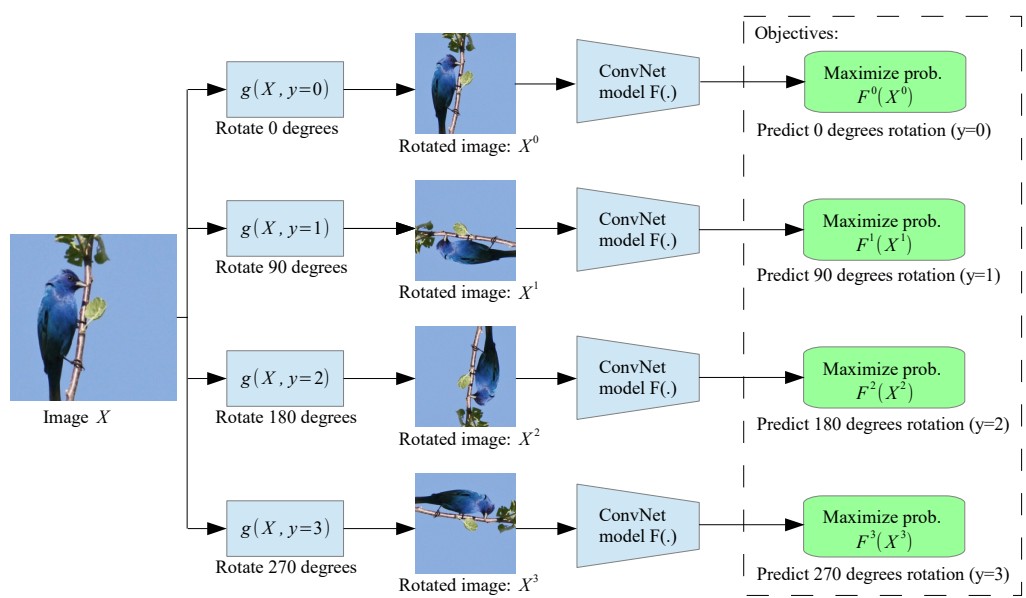

Figure 2: Illustration of the self-supervised task that we propose for semantic feature learning. Given four possible geometric transformations, the 0, 90, 180, and 270 degrees rotations, we train a ConvNet model $F(.)$ to recognize the rotation that is applied to the image that it gets as input. $F^y(X^{y^*})$ is the probability of rotation transformation $y$ predicted by model $F(.)$ when it gets as input an image that has been transformed by the rotation transformation $y^*$.

to successfully predict the rotation of an image the ConvNet model must necessarily learn to localize salient objects in the image, recognize their orientation and object type, and then relate the object orientation with the dominant orientation that each type of object tends to be depicted within the available images. In Figure 3b we visualize some attention maps generated by a model trained on the rotation recognition task. These attention maps are computed based on the magnitude of activations at each spatial cell of a convolutional layer and essentially reflect where the network puts most of its focus in order to classify an input image. We observe, indeed, that in order for the model to accomplish the rotation prediction task it learns to focus on high level object parts in the image, such as eyes, nose, tails, and heads. By comparing them with the attention maps generated by a model trained on the object recognition task in a supervised way (see Figure 3a) we observe that both models seem to focus on roughly the same image regions. Furthermore, in Figure 4 we visualize the first layer filters that were learnt by an AlexNet model trained on the proposed rotation recognition task. As can be seen, they appear to have a big variety of edge filters on multiple orientations and multiple frequencies. Remarkably, these filters seem to have a greater amount of variety even than the filters learnt by the supervised object recognition task.

**Absence of low-level visual artifacts:** An additional important advantage of using image rotations by multiples of 90 degrees over other geometric transformations, is that they can be implemented by flip and transpose operations (as we will see below) that do not leave any easily detectable low-level visual artifacts that will lead the ConvNet to learn trivial features with no practical value for the vision perception tasks. In contrast, had we decided to use as geometric transformations, e.g., scale and aspect ratio image transformations, in order to implement them we would need to use image resizing routines that leave easily detectable image artifacts.

**Well-posedness:** Furthermore, human captured images tend to depict objects in an "up-standing" position, thus making the rotation recognition task well defined, i.e., given an image rotated by 0, 90, 180, or 270 degrees, there is usually no ambiguity of what is the rotation transformation (with the exception of images that only depict round objects). In contrast, that is not the case for the object scale that varies significantly on human captured images.

**Implementing image rotations:** In order to implement the image rotations by 90, 180, and 270 degrees (the 0 degrees case is the image itself), we use flip and transpose operations. Specifically,

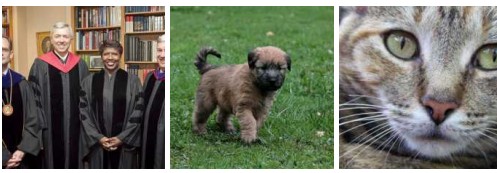

Input images on the models

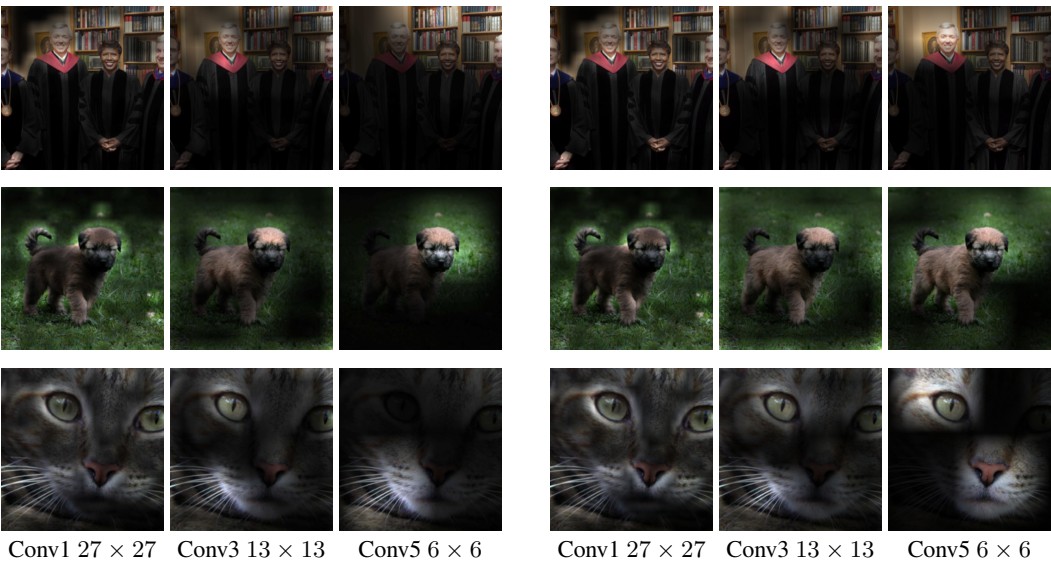

| Conv1 $27 \times 27$ | Conv3 $13 \times 13$ | Conv5 $6 \times 6$ | Conv1 $27 \times 27$ | Conv3 $13 \times 13$ | Conv5 $6 \times 6$ |

(a) **Attention maps of supervised model**     (b) **Attention maps of our self-supervised model**

Figure 3: Attention maps generated by an AlexNet model trained **(a)** to recognize objects (supervised), and **(b)** to recognize image rotations (self-supervised). In order to generate the attention map of a conv. layer we first compute the feature maps of this layer, then we raise each feature activation on the power $p$, and finally we sum the activations at each location of the feature map. For the conv. layers 1, 2, and 3 we used the powers $p = 1$, $p = 2$, and $p = 4$ respectively. For visualization of our self-supervised model's attention maps for all the rotated versions of the images see Figure 6 in appendix A.

for 90 degrees rotation we first transpose the image and then flip it vertically (upside-down flip), for 180 degrees rotation we flip the image first vertically and then horizontally (left-right flip), and finally for 270 degrees rotation we first flip vertically the image and then we transpose it.

## 2.3 DISCUSSION

The simple formulation of our self-supervised task has several advantages. It has the same computational cost as supervised learning, similar training convergence speed (that is significantly faster than image reconstruction based approaches; our AlexNet model trains in around 2 days using a single Titan X GPU), and can trivially adopt the efficient parallelization schemes devised for supervised learning (Goyal et al., 2017), making it an ideal candidate for unsupervised learning on internet-scale data (i.e., billions of images). Furthermore, our approach does not require any special image pre-processing routine in order to avoid learning trivial features, as many other unsupervised or self-supervised approaches do. Despite the simplicity of our self-supervised formulation, as we will see in the experimental section of the paper, the features learned by our approach achieve dramatic improvements on the unsupervised feature learning benchmarks.

## 3 EXPERIMENTAL RESULTS

In this section we conduct an extensive evaluation of our approach on the most commonly used image datasets, such as CIFAR-10 (Krizhevsky & Hinton, 2009), ImageNet (Russakovsky et al., 2015),

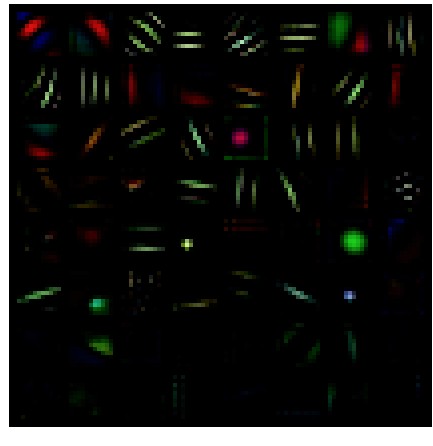 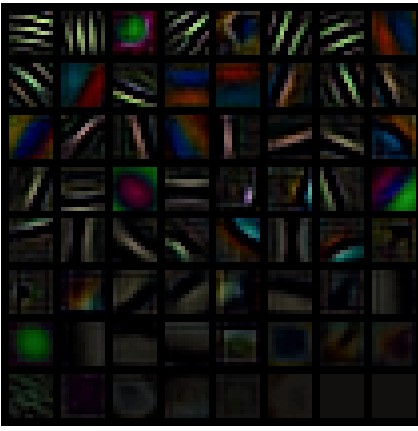

(a) **Supervised**          (b) **Self-supervised to recognize rotations**

Figure 4: First layer filters learned by a AlexNet model trained on **(a)** the supervised object recognition task and **(b)** the self-supervised task of recognizing rotated images. We observe that the filters learned by the self-supervised task are mostly oriented edge filters on various frequencies and, remarkably, they seem to have more variety than those learned on the supervised task.

Table 1: Evaluation of the unsupervised learned features by measuring the classification accuracy that they achieve when we train a non-linear object classifier on top of them. The reported results are from CIFAR-10. The size of the ConvB1 feature maps is $96 \times 16 \times 16$ and the size of the rest feature maps is $192 \times 8 \times 8$.

| Model | ConvB1 | ConvB2 | ConvB3 | ConvB4 | ConvB5 |
|---|---|---|---|---|---|
| RotNet with 3 conv. blocks | 85.45 | 88.26 | 62.09 | - | - |
| RotNet with 4 conv. blocks | 85.07 | 89.06 | 86.21 | 61.73 | - |
| RotNet with 5 conv. blocks | 85.04 | **89.76** | 86.82 | 74.50 | 50.37 |

PASCAL (Everingham et al., 2010), and Places205 (Zhou et al., 2014), as well as on various vision tasks, such as object detection, object segmentation, and image classification. We also consider several learning scenarios, including transfer learning and semi-supervised learning. In all cases, we compare our approach with corresponding state-of-the-art methods.

### 3.1 CIFAR EXPERIMENTS

We start by evaluating on the object recognition task of CIFAR-10 the ConvNet based features learned by the proposed self-supervised task of rotation recognition. We will here after call a ConvNet model that is trained on the self-supervised task of rotation recognition *RotNet* model.

**Implementation details:** In our CIFAR-10 experiments we implement the *RotNet* models with Network-In-Network (NIN) architectures (Lin et al., 2013). In order to train them on the rotation prediction task, we use SGD with batch size 128, momentum 0.9, weight decay $5e - 4$ and $lr$ of 0.1. We drop the learning rates by a factor of 5 after epochs 30, 60, and 80. We train in total for 100 epochs. In our preliminary experiments we found that we get significant improvement when during training we train the network by feeding it all the four rotated copies of an image simultaneously instead of each time randomly sampling a single rotation transformation. Therefore, at each training batch the network sees 4 times more images than the batch size.

**Evaluation of the learned feature hierarchies:** First, we explore how the quality of the learned features depends from their depth (i.e., the depth of the layer that they come from) as well as from the total depth of the *RotNet* model. For that purpose, we first train using the CIFAR-10 training images three *RotNet* models which have 3, 4, and 5 convolutional blocks respectively (note that each conv. block in the NIN architectures that implement our *RotNet* models have 3 conv. layers; therefore,

Table 2: Exploring the quality of the self-supervised learned features w.r.t. the number of recognized rotations. For all the entries we trained a non-linear classifier with 3 fully connected layers (similar to Table 1) on top of the feature maps generated by the 2nd conv. block of a RotNet model with 4 conv. blocks in total. The reported results are from CIFAR-10.

| # Rotations | Rotations | CIFAR-10 Classification Accuracy |
|---|---|---|
| 4 | 0°, 90°, 180°, 270° | **89.06** |
| 8 | 0°, 45°, 90°, 135°, 180°, 225°, 270°, 315° | 88.51 |
| 2 | 0°, 180° | 87.46 |
| 2 | 90°, 270° | 85.52 |

Table 3: Evaluation of unsupervised feature learning methods on CIFAR-10. The *Supervised NIN* and the *(Ours) RotNet + conv* entries have exactly the same architecture but the first was trained fully supervised while on the second the first 2 conv. blocks were trained unsupervised with our rotation prediction task and the 3rd block only was trained in a supervised manner. In the *Random Init. + conv* entry a conv. classifier (similar to that of *(Ours) RotNet + conv*) is trained on top of two NIN conv. blocks that are randomly initialized and stay frozen. Note that each of the prior approaches has a different ConvNet architecture and thus the comparison with them is just indicative.

| Method | Accuracy |
|---|---|
| Supervised NIN | 92.80 |
| Random Init. + conv | 72.50 |
| (Ours) RotNet + non-linear | 89.06 |
| (Ours) RotNet + conv | **91.16** |
| (Ours) RotNet + non-linear (fine-tuned) | 91.73 |
| (Ours) RotNet + conv (fine-tuned) | 92.17 |
| Roto-Scat + SVM Oyallon & Mallat (2015) | 82.3 |
| ExemplarCNN Dosovitskiy et al. (2014) | 84.3 |
| DCGAN Radford et al. (2015) | 82.8 |
| Scattering Oyallon et al. (2017) | 84.7 |

the total number of conv. layers of the examined *RotNet* models is 9, 12, and 15 for 3, 4, and 5 conv. blocks respectively). Afterwards, we learn classifiers on top of the feature maps generated by each conv. block of each *RotNet* model. Those classifiers are trained in a supervised way on the object recognition task of CIFAR-10. They consist of 3 fully connected layers; the 2 hidden layers have 200 feature channels each and are followed by batch-norm and relu units. We report the accuracy results of CIFAR-10 test set in Table 1. We observe that in all cases the feature maps generated by the 2nd conv. block (that actually has depth 6 in terms of the total number of conv. layer till that point) achieve the highest accuracy, i.e., between 88.26% and 89.06%. The features of the conv. blocks that follow the 2nd one gradually degrade the object recognition accuracy, which we assume is because they start becoming more and more specific on the self-supervised task of rotation prediction. Also, we observe that increasing the total depth of the RotNet models leads to increased object recognition performance by the feature maps generated by earlier layers (and after the 1st conv. block). We assume that this is because increasing the depth of the model and thus the complexity of its head (i.e., top ConvNet layers) allows the features of earlier layers to be less specific to the rotation prediction task.

**Exploring the quality of the learned features w.r.t. the number of recognized rotations:** In Table 2 we explore how the quality of the self-supervised features depends on the number of discrete rotations used in the rotation prediction task. For that purpose we defined three extra rotation recognition tasks: (a) one with 8 rotations that includes all the multiples of 45 degrees, (b) one with only the 0° and 180° rotations, and (c) one with only the 90° and 270° rotations. In order to implement the rotation transformation of the 45°, 135°, 225°, 270°, and 315° rotations (in the 8 discrete rotations case) we used an image wrapping routine and then we took care to crop only the central square

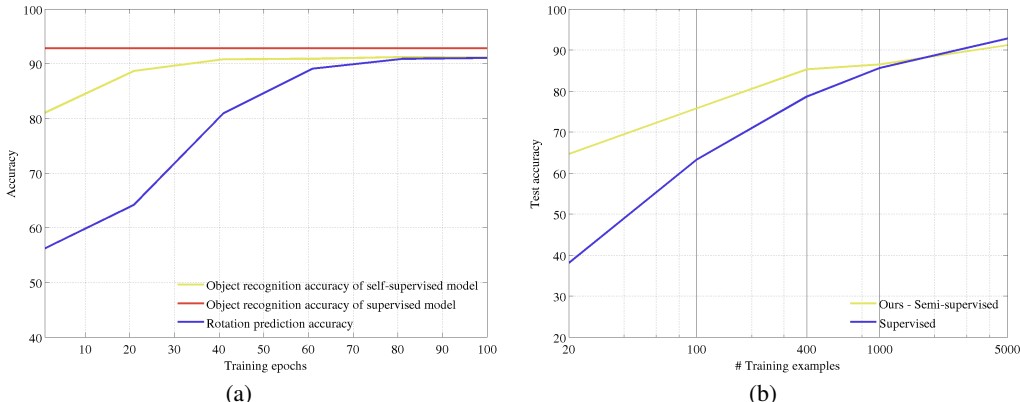

Figure 5: **(a)** Plot with the rotation prediction accuracy and object recognition accuracy as a function of the training epochs used for solving the rotation prediction task. The red curve is the object recognition accuracy of a fully supervised model (a NIN model), which is independent from the training epochs on the rotation prediction task. The yellow curve is the object recognition accuracy of an object classifier trained on top of feature maps learned by a *RotNet* model at different snapshots of the training procedure. **(b)** Accuracy as a function of the number of training examples per category in CIFAR-10. *Ours semi-supervised* is a NIN model that the first 2 conv. blocks are *RotNet* model that was trained in a self-supervised way on the entire training set of CIFAR-10 and the 3rd conv. block along with a prediction linear layer that was trained with the object recognition task only on the available set of labeled images.

image regions that do not include any of the empty image areas introduced by the rotation transformations (and which can easily indicate the image rotation). We observe that indeed for 4 discrete rotations (as we proposed) we achieve better object recognition performance than the 8 or 2 cases. We believe that this is because the 2 orientations case offers too few classes for recognition (i.e., less supervisory information is provided) while in the 8 orientations case the geometric transformations are not distinguishable enough and furthermore the 4 extra rotations introduced may lead to visual artifacts on the rotated images. Moreover, we observe that among the RotNet models trained with 2 discrete rotations, the RotNet model trained with $90°$ and $270°$ rotations achieves worse object recognition performance than the model trained with the $0°$ and $180°$ rotations, which is probably due to the fact that the former model does not "see" during the unsupervised phase the $0°$ rotation that is typically used during the object recognition training phase.

**Comparison against supervised and other unsupervised methods:** In Table 3 we compare our unsupervised learned features against other unsupervised (or hand-crafted) features on CIFAR-10. For our entries we use the feature maps generated by the 2nd conv. block of a RotNet model with 4 conv. blocks in total. On top of those RotNet features we train 2 different classifiers: (a) a non-linear classifier with 3 fully connected layers as before (entry *(Ours) RotNet + non-linear*), and (b) three conv. layers plus a linear prediction layer (entry *(Ours) RotNet +conv.*; note that this entry is basically a 3 blocks NIN model with the first 2 blocks coming from a RotNet model and the 3rd being randomly initialized and trained on the recognition task). We observe that we improve over the prior unsupervised approaches and we achieve state-of-the-art results in CIFAR-10 (note that each of the prior approaches has a different ConvNet architecture thus the comparison with them is just indicative). More notably, *the accuracy gap between the RotNet based model and the fully supervised NIN model is very small*, only 1.64 percentage points (92.80% vs 91.16%). We provide per class breakdown of the classification accuracy of our unsupervised model as well as the supervised one in Table 9 (in appendix B). In Table 3 we also report the performance of the RotNet features when, instead of being kept frozen, they are fine-tuned during the object recognition training phase. We observe that fine-tuning the unsupervised learned features further improves the classification performance, thus reducing even more the gap with the supervised case.

**Correlation between object classification task and rotation prediction task:** In Figure 5a, we plot the object classification accuracy as a function of the training epochs used for solving the self-supervised task of recognizing rotations, which learns the features used by the object classifier.

Table 4: **Task Generalization: ImageNet top-1 classification with non-linear layers**. We compare our unsupervised feature learning approach with other unsupervised approaches by training non-linear classifiers on top of the feature maps of each layer to perform the 1000-way ImageNet classification task, as proposed by Noroozi & Favaro (2016). For instance, for the conv5 feature map we train the layers that follow the conv5 layer in the AlexNet architecture (i.e., fc6, fc7, and fc8). Similarly for the conv4 feature maps. We implemented those non-linear classifiers with batch normalization units after each linear layer (fully connected or convolutional) and without employing drop out units. All approaches use AlexNet variants and were pre-trained on ImageNet without labels except the ImageNet labels and Random entries. During testing we use a single crop and do not perform flipping augmentation. We report top-1 classification accuracy.

| Method | Conv4 | Conv5 |
|---|---|---|
| ImageNet labels from (Bojanowski & Joulin, 2017) | 59.7 | 59.7 |
| Random from (Noroozi & Favaro, 2016) | 27.1 | 12.0 |
| Tracking Wang & Gupta (2015) | 38.8 | 29.8 |
| Context (Doersch et al., 2015) | 45.6 | 30.4 |
| Colorization (Zhang et al., 2016a) | 40.7 | 35.2 |
| Jigsaw Puzzles (Noroozi & Favaro, 2016) | 45.3 | 34.6 |
| BIGAN (Donahue et al., 2016) | 41.9 | 32.2 |
| NAT (Bojanowski & Joulin, 2017) | - | 36.0 |
| (Ours) RotNet | 50.0 | 43.8 |

More specifically, in order to create the object recognition accuracy curve, in each training snapshot of RotNet (i.e., every 20 epochs), we pause its training procedure and we train from scratch (until convergence) a non-linear object classifier on top of the so far learnt RotNet features. Therefore, the object recognition accuracy curve depicts the accuracy of those non-linear object classifiers after the end of their training while the rotation prediction accuracy curve depicts the accuracy of the RotNet at those snapshots. We observe that, as the ability of the RotNet features for solving the rotation prediction task improves (i.e., as the rotation prediction accuracy increases), their ability to help solving the object recognition task improves as well (i.e., the object recognition accuracy also increases). Furthermore, we observe that the object recognition accuracy converges fast w.r.t. the number of training epochs used for solving the pretext task of rotation prediction.

**Semi-supervised setting:** Motivated by the very high performance of our unsupervised feature learning method, we also evaluate it on a semi-supervised setting. More specifically, we first train a 4 block *RotNet* model on the rotation prediction task using the entire image dataset of CIFAR-10 and then we train on top of its feature maps object classifiers using only a subset of the available images and their corresponding labels. As feature maps we use those generated by the 2nd conv. block of the *RotNet* model. As a classifier we use a set of convolutional layers that actually has the same architecture as the 3rd conv. block of a NIN model plus a linear classifier, all randomly initialized. For training the object classifier we use for each category 20, 100, 400, 1000, or 5000 image examples. Note that 5000 image examples is the extreme case of using the entire CIFAR-10 training dataset. Also, we compare our method with a supervised model that is trained only on the available examples each time. In Figure 5b we plot the accuracy of the examined models as a function of the available training examples. We observe that our unsupervised trained model exceeds in this semi-supervised setting the supervised model when the number of examples per category drops below 1000. Furthermore, as the number of examples decreases, the performance gap in favor of our method is increased. This empirical evidence demonstrates the usefulness of our method on semi-supervised settings.

## 3.2 EVALUATION OF SELF-SUPERVISED FEATURES TRAINED IN IMAGENET

Here we evaluate the performance of our self-supervised ConvNet models on the ImageNet, Places, and PASCAL VOC datasets. Specifically, we first train a *RotNet* model on the training images of the ImageNet dataset and then we evaluate the performance of the self-supervised features on the image

Table 5: **Task Generalization: ImageNet top-1 classification with linear layers**. We compare our unsupervised feature learning approach with other unsupervised approaches by training logistic regression classifiers on top of the feature maps of each layer to perform the 1000-way ImageNet classification task, as proposed by Zhang et al. (2016a). All weights are frozen and feature maps are spatially resized (with adaptive max pooling) so as to have around 9000 elements. All approaches use AlexNet variants and were pre-trained on ImageNet without labels except the ImageNet labels and Random entries.

| Method | Conv1 | Conv2 | Conv3 | Conv4 | Conv5 |
|---|---|---|---|---|---|
| ImageNet labels | 19.3 | 36.3 | 44.2 | 48.3 | 50.5 |
| Random | 11.6 | 17.1 | 16.9 | 16.3 | 14.1 |
| Random rescaled Krähenbühl et al. (2015) | 17.5 | 23.0 | 24.5 | 23.2 | 20.6 |
| Context (Doersch et al., 2015) | 16.2 | 23.3 | 30.2 | 31.7 | 29.6 |
| Context Encoders (Pathak et al., 2016b) | 14.1 | 20.7 | 21.0 | 19.8 | 15.5 |
| Colorization (Zhang et al., 2016a) | 12.5 | 24.5 | 30.4 | 31.5 | 30.3 |
| Jigsaw Puzzles (Noroozi & Favaro, 2016) | 18.2 | 28.8 | 34.0 | 33.9 | 27.1 |
| BIGAN (Donahue et al., 2016) | 17.7 | 24.5 | 31.0 | 29.9 | 28.0 |
| Split-Brain (Zhang et al., 2016b) | 17.7 | 29.3 | 35.4 | 35.2 | 32.8 |
| Counting (Noroozi et al., 2017) | 18.0 | 30.6 | 34.3 | 32.5 | 25.7 |
| (Ours) RotNet | **18.8** | **31.7** | **38.7** | **38.2** | **36.5** |

Table 6: **Task & Dataset Generalization: Places top-1 classification with linear layers**. We compare our unsupervised feature learning approach with other unsupervised approaches by training logistic regression classifiers on top of the feature maps of each layer to perform the 205-way Places classification task (Zhou et al., 2014). All unsupervised methods are pre-trained (in an unsupervised way) on ImageNet. All weights are frozen and feature maps are spatially resized (with adaptive max pooling) so as to have around 9000 elements. All approaches use AlexNet variants and were pre-trained on ImageNet without labels except the Place labels, ImageNet labels, and Random entries.

| Method | Conv1 | Conv2 | Conv3 | Conv4 | Conv5 |
|---|---|---|---|---|---|
| Places labels Zhou et al. (2014) | 22.1 | 35.1 | 40.2 | 43.3 | 44.6 |
| ImageNet labels | 22.7 | 34.8 | 38.4 | 39.4 | 38.7 |
| Random | 15.7 | 20.3 | 19.8 | 19.1 | 17.5 |
| Random rescaled Krähenbühl et al. (2015) | 21.4 | 26.2 | 27.1 | 26.1 | 24.0 |
| Context (Doersch et al., 2015) | 19.7 | 26.7 | 31.9 | 32.7 | 30.9 |
| Context Encoders (Pathak et al., 2016b) | 18.2 | 23.2 | 23.4 | 21.9 | 18.4 |
| Colorization (Zhang et al., 2016a) | 16.0 | 25.7 | 29.6 | 30.3 | 29.7 |
| Jigsaw Puzzles (Noroozi & Favaro, 2016) | 23.0 | 31.9 | 35.0 | 34.2 | 29.3 |
| BIGAN (Donahue et al., 2016) | 22.0 | 28.7 | 31.8 | 31.3 | 29.7 |
| Split-Brain (Zhang et al., 2016b) | 21.3 | 30.7 | 34.0 | 34.1 | 32.5 |
| Counting (Noroozi et al., 2017) | **23.3** | **33.9** | **36.3** | **34.7** | 29.6 |
| (Ours) RotNet | 21.5 | 31.0 | 35.1 | 34.6 | **33.7** |

classification tasks of ImageNet, Places, and PASCAL VOC datasets and on the object detection and object segmentation tasks of PASCAL VOC.

**Implementation details:** For those experiments we implemented our *RotNet* model with an AlexNet architecture. Our implementation of the AlexNet model does not have local response normalization units, dropout units, or groups in the colvolutional layers while it includes batch normalization units after each linear layer (either convolutional or fully connected). In order to train the AlexNet based *RotNet* model, we use SGD with batch size 192, momentum 0.9, weight decay $5e - 4$ and $lr$ of 0.01. We drop the learning rates by a factor of 10 after epochs 10, and 20 epochs. We train in total for 30 epochs. As in the CIFAR experiments, during training we feed the *RotNet* model all four rotated copies of an image simultaneously (in the same mini-batch).

Table 7: **Task & Dataset Generalization: PASCAL VOC 2007 classification and detection results, and PASCAL VOC 2012 segmentation results.** We used the publicly available testing frameworks of Krähenbühl et al. (2015) for classification, of Girshick (2015) for detection, and of Long et al. (2015) for segmentation. For classification, we either fix the features before conv5 (column *fc6-8*) or we fine-tune the whole model (column *all*). For detection we use multi-scale training and single scale testing. All approaches use AlexNet variants and were pre-trained on ImageNet without labels except the ImageNet labels and Random entries. After unsupervised training, we absorb the batch normalization units on the linear layers and we use the weight rescaling technique proposed by Krähenbühl et al. (2015) (which is common among the unsupervised methods). As customary, we report the mean average precision (mAP) on the classification and detection tasks, and the mean intersection over union (mIoU) on the segmentation task.

| | Classification (%mAP) | | Detection (%mAP) | Segmentation (%mIoU) |
|---|---|---|---|---|
| Trained layers | fc6-8 | all | all | all |
| ImageNet labels | 78.9 | 79.9 | 56.8 | 48.0 |
| Random | | 53.3 | 43.4 | 19.8 |
| Random rescaled Krähenbühl et al. (2015) | 39.2 | 56.6 | 45.6 | 32.6 |
| Egomotion (Agrawal et al., 2015) | 31.0 | 54.2 | 43.9 | |
| Context Encoders (Pathak et al., 2016b) | 34.6 | 56.5 | 44.5 | 29.7 |
| Tracking (Wang & Gupta, 2015) | 55.6 | 63.1 | 47.4 | |
| Context (Doersch et al., 2015) | 55.1 | 65.3 | 51.1 | |
| Colorization (Zhang et al., 2016a) | 61.5 | 65.6 | 46.9 | 35.6 |
| BIGAN (Donahue et al., 2016) | 52.3 | 60.1 | 46.9 | 34.9 |
| Jigsaw Puzzles (Noroozi & Favaro, 2016) | - | 67.6 | 53.2 | 37.6 |
| NAT (Bojanowski & Joulin, 2017) | 56.7 | 65.3 | 49.4 | |
| Split-Brain (Zhang et al., 2016b) | 63.0 | 67.1 | 46.7 | 36.0 |
| ColorProxy (Larsson et al., 2017) | | 65.9 | | 38.4 |
| Counting (Noroozi et al., 2017) | - | 67.7 | 51.4 | 36.6 |
| **(Ours) RotNet** | **70.87** | **72.97** | **54.4** | **39.1** |

**ImageNet classification task:** We evaluate the task generalization of our self-supervised learned features by training on top of them non-linear object classifiers for the ImageNet classification task (following the evaluation scheme of (Noroozi & Favaro, 2016)). In Table 4 we report the classification performance of our self-supervised features and we compare it with the other unsupervised approaches. *We observe that our approach surpasses all the other methods by a significant margin.* For the feature maps generated by the Conv4 layer, our improvement is more than 4 percentage points and for the feature maps generated by the Conv5 layer, our improvement is even bigger, around 8 percentage points. Furthermore, our approach significantly narrows the performance gap between unsupervised features and supervised features. In Table 5 we report similar results but for linear (logistic regression) classifiers (following the evaluation scheme of Zhang et al. (2016a)). Again, our unsupervised method demonstrates significant improvements over prior unsupervised methods.

**Transfer learning evaluation on PASCAL VOC:** In Table 7 we evaluate the task and dataset generalization of our unsupervised learned features by fine-tuning them on the PASCAL VOC classification, detection, and segmentation tasks. As with the ImageNet classification task, we outperform by significant margin all the competing unsupervised methods in all tested tasks, significantly narrowing the gap with the supervised case. Notably, the PASCAL VOC 2007 object detection performance that our self-supervised model achieves is $54.4\%$ mAP, which is only 2.4 points lower than the supervised case. We provide the per class detection performance of our method in Table 8 (in appendix B).

**Places classification task:** In Table 6 we evaluate the task and dataset generalization of our approach by training linear (logistic regression) classifiers on top of the learned features in order to perform the 205-way Places classification task. Note that in this case the learnt features are evaluated w.r.t.

their generalization on classes that were "unseen" during the unsupervised training phase. As can be seen, even in this case our method manages to either surpass or achieve comparable results w.r.t. prior state-of-the-art unsupervised learning approaches.

## 4 CONCLUSIONS

In our work we propose a novel formulation for self-supervised feature learning that trains a ConvNet model to be able to recognize the image rotation that has been applied to its input images. Despite the simplicity of our self-supervised task, we demonstrate that it successfully forces the ConvNet model trained on it to learn semantic features that are useful for a variety of visual perception tasks, such as object recognition, object detection, and object segmentation. We exhaustively evaluate our method in various unsupervised and semi-supervised benchmarks and we achieve in all of them state-of-the-art performance. Specifically, our self-supervised approach manages to drastically improve the state-of-the-art results on unsupervised feature learning for ImageNet classification, PASCAL classification, PASCAL detection, PASCAL segmentation, and CIFAR-10 classification, surpassing prior approaches by a significant margin and thus drastically reducing the gap between unsupervised and supervised feature learning.

## 5 ACKNOWLEDGEMENTS

This work was supported by the ANR SEMAPOLIS project, an INTEL gift, and hardware donation by NVIDIA.

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

## APPENDIX A    VISUALIZING ATTENTION MAPS OF ROTATED IMAGES

Here we visualize the attention maps generated by an AlexNet model trained on the self-supervised task of rotation recognition for all the rotated copies of a few images. We observe that the attention maps of all the rotated copies of an image are roughly the same, i.e., the attention maps are equivariant w.r.t. the image rotations. This practically means that in order to accomplish the rotation prediction task the network focuses on the same object parts regardless of the image rotation.

**Attention maps of Conv3 feature maps (size: $13 \times 13$)**

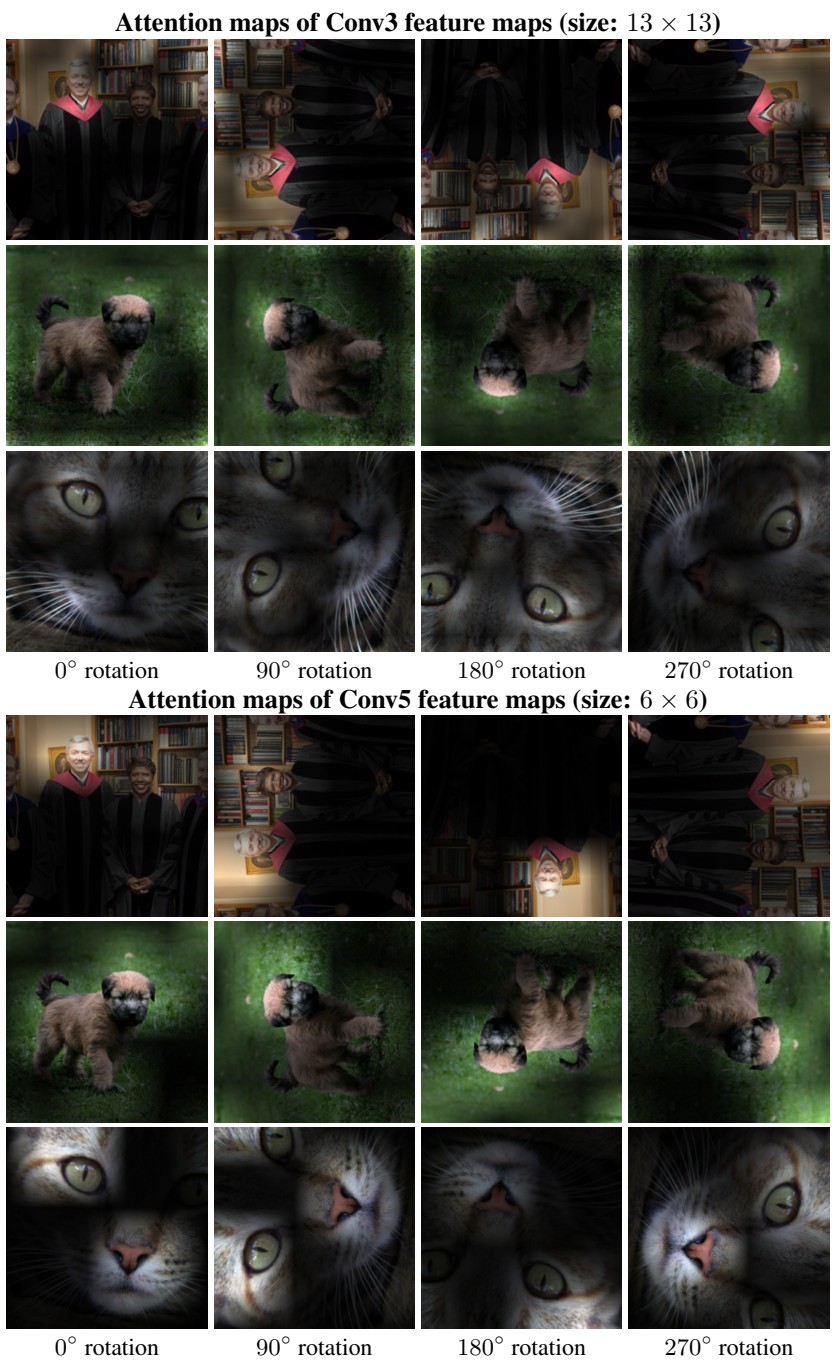

Figure 6: Attention maps of the Conv3 and Conv5 feature maps generated by an AlexNet model trained on the self-supervised task of recognizing image rotations. Here we present the attention maps generated for all the 4 rotated copies of an image.

## APPENDIX B  PER CLASS BREAKDOWN OF DETECTION AND CLASSIFICATION PERFORMANCE

In Tables 8 and 9 we report the per class performance of our unsupervised learning method on the PASCAL detection and CIFAR-10 classification tasks respectively.

Table 8: **Per class PASCAL VOC 2007 detection performance.** As usual, we report the average precision metric. The results of the supervised model (i.e., ImageNet labels entry) come from Doersch et al. (2015).

| Classes | aero | bike | bird | boat | bottle | bus | car | cat | chair | cow | table | dog | horse | mbike | person | plant | sheep | sofa | train | tv |
|---|---|---|---|---|---|---|---|---|---|---|---|---|---|---|---|---|---|---|---|---|
| ImageNet labels | 64.0 | **69.6** | **53.2** | **44.4** | **24.9** | **65.7** | **69.6** | **69.2** | 28.9 | **63.6** | **62.8** | **63.9** | **73.3** | 64.6 | 55.8 | **25.7** | **50.5** | 55.4 | 69.3 | **56.4** |
| (Ours) RotNet | **65.5** | 65.3 | 43.8 | 39.8 | 20.2 | 65.4 | 69.2 | 63.9 | **30.2** | 56.3 | 62.3 | 56.8 | 71.6 | **67.2** | **56.3** | 22.7 | 45.6 | **59.5** | **71.6** | 55.3 |

Table 9: **Per class CIFAR-10 classification accuracy.**

| Classes | aero | car | bird | cat | deer | dog | frog | horse | ship | truck |
|---|---|---|---|---|---|---|---|---|---|---|
| Supervised | **93.7** | **96.3** | **89.4** | 82.4 | **93.6** | **89.7** | **95.0** | **94.3** | **95.7** | **95.2** |
| (Ours) RotNet | 91.7 | 95.8 | 87.1 | **83.5** | 91.5 | 85.3 | 94.2 | 91.9 | **95.7** | 94.2 |

