# OpenReview forum: "Unsupervised Representation Learning by Predicting Image Rotations"
_ICLR.cc/2018/Conference — Accept (Poster)_

### Official Review · AnonReviewer1 · 2017-11-25
**Interesting discovery, good results, but not a lot of content.**

**Rating:** 6
**Confidence:** 5

**Review:**

The paper proposes a simple classification task for learning feature extractors without requiring manual annotations: predicting one of four rotations that the image has been subjected to: by 0, 90, 180 or 270º. Then the paper shows that pre-training on this task leads to state-of-the-art results on a number of popular benchmarks for object recognition, when training classifiers on top of the resulting representation.

This is a useful discovery, because generating the rotated images is trivial to implement by anyone. It is a special case of the approach by Agrawal et al 2015, with more efficiency.

On the negative side, this line of work would benefit from demonstrating concrete benefits. The performance obtained by pre-training with rotations is still inferior to performance obtained by pre-training with ImageNet, and we do have ImageNet so there is no reason not to use it. It would be important to come up with tasks for which there is not one ImageNet, then techniques such as that proposed in the paper would be necessary. However rotations are somewhat specific to images. There may be opportunities with some type of medical data.

Additionally, the scope of the paper is a little bit restricted, there is not that much to take home besides the the following information: "predicting rotations seems to require a lot of object category recognition".

---

> ### Author Response · Authors · 2017-12-24
> **Response to AnonReviewer1**
>
> We thank the reviewer for the valuable feedback.  Here we will answer to his/her comments.
>
> Comment:
> "This is a useful discovery, because generating the rotated images is trivial to implement by anyone. It is a special case of the approach by Agrawal et al 2015, with more efficiency."
>
> -----
>
> Answer:
> We would like to mention that our method is NOT a special case of the Egomotion method of Agrawal et al 2015. More specifically, the Egomotion method employs a ConvNet model with siamese like architecture that takes as input TWO CONSECUTIVE VIDEO FRAMES and is trained to predict (through regression) their camera transformation. Instead, in our approach, the ConvNet takes as input A SINGLE IMAGE to which we have applied a random geometric transformation (rotation) and is trained to recognize (through classification) this geometric transformation WITHOUT HAVING ACCESS TO THE INITIAL IMAGE. These are two fundamentally different approaches.
>
> To make this difference more clear, consider, for instance, the case where we feed the siamese ConvNet model of the Egomotion method with a pair of images that include an image and a rotated copy of it. In this case, training the ConvNet to predict the difference in rotation would lead to learning some very trivial features (e.g., it would only need to compare the four corners of the image in order to solve the task).
>
> On the contrary, since the ConvNet used in our approach takes as input a single image, it cannot recognize the type of geometric transformation applied between consecutive frames and used in the Egomotion method. Instead, our approach requires utilizing geometric transformations (e.g., the image rotations of 0, 90, 180, and 270 degrees) that transform the image in such a manner that is unambiguous to infer the applied transformation without having access to the initial image.
>
> We believe that the above differences force our ConvNet model to learn different and, in our opinion, more high level features from the Egomotion method.
>
> Comments:
> "On the negative side, this line of work would benefit from demonstrating concrete benefits. The performance obtained by pre-training with rotations is still inferior to performance obtained by pre-training with ImageNet, and we do have ImageNet so there is no reason not to use it. It would be important to come up with tasks for which there is not one ImageNet, then techniques such as that proposed in the paper would be necessary. However rotations are somewhat specific to images. There may be opportunities with some type of medical data."
>
> "Additionally, the scope of the paper is a little bit restricted, there is not that much to take home besides the the following information:  predicting rotations seems to require a lot of object category recognition."
>
> -----
>
> Answer:
> To be honest, we found the above criticism of our method unfair. It is true that one would ultimately like the performance of an unsupervised learning approach to surpass the performance of a fully supervised one, but, to the best of our knowledge, no published unsupervised learning method manages to achieve this goal so far.
>
> Furthermore, the type of data our work focuses on (i.e., visual data) have attracted a tremendous amount of research interest as far as unsupervised learning is concernced. In fact, just over the last few years, one can cite numerous other unsupervised learning works focusing on exactly the same type of data (e.g., [1,2,3,4,5]), all of which have been very well received by the community and which have also appeared on top-rank computer vision or machine learning conferences.
>
> [1] Learning to See by Moving
> [2] Unsupervised Visual Representation Learning by Context Prediction
> [3] Colorful Image Colorization.
> [4] Unsupervised learning of visual representations by solving jigsaw puzzles.
> [5] Representation Learning by Learning to Count

---

### Official Review · AnonReviewer3 · 2017-11-27
**remarkably simple but effective strategy, some missing experiments, awkward writing**

**Rating:** 6
**Confidence:** 4

**Review:**

Strengths:
* Very simple strategy for unsupervised learning of deep image features. Simplicity of approach is a good quality in my view.
* The rationale for the effectiveness of the approach is explained well.
* The representation learned from unlabeled data is shown to yield strong results on image categorization (albeit mostly in scenarios where the unsupervised features have been learned from the *same* dataset where classification is performed -- more on this below).
* The image rotations are implemented in terms of flipping and transposition, which do not create visual artifacts easily recognizable by deep models.

Weaknesses:
* There are several obvious additional experiments that, in my view, would greatly strengthen this work:
1. Nearly all of the image categorization results (with the exception of those in Table 4) are presented for the contrived scenario where the unsupervised representation is learned from the same training set as the one used for the final supervised training of the categorization model. This is a useless application scenario. If labels for the training examples are available, why not using them for feature learning given that this leads to improved performance (see results in Tables)? More importantly, this setup does not allow us to understand how general the unsupervised features are. Maybe they are effective  precisely because they have been learned from images of the 10 classes that the final classifier needs to distinguish... I would have liked to see some results involving unsupervised learning from a dataset that may contain classes different from those of the final test classification or, even better, from a dataset of randomly selected images that lack categorical coherence (e.g., photos randomly picked from the Web, such as Flickr pics).
2. In nearly all the experiments, the classifier is built on top of the frozen unsupervised features. This is in contrast with the common practice of finetuning the entire pretrained unsupervised net on the supervised task. It'd be good to know why the authors opted for the different setup and to see in any case some supervised finetuning results.
3. It would be useful to see the accuracy per class both when using unsupervised features as well as fully-supervised features. There are many objects that have a canonical pose/rotation in the world. Forcing the unsupervised features to distinguish rotations of such objects may affect the recognition accuracy for these classes. Thus, my request for seeing how the unsupervised learning affects class-specific accuracy.
4. While the results in Table 2 are impressive, it appears that the different unsupervised learning methods reported in this table are based on different architectures. This raises the question of whether performance gains are due to the better mechanism for unsupervised learning or rather the better network architecture.
5. I do understand that using only 0, 90, 180 and 270 degree rotations eliminates the issue of potentially recognizable artifacts. Nevertheless, it'd be interesting to see what happens empirically when the number of discrete rotations is increased, e.g., by including 45, 135, 225 and 315 degree rotations. And what happens if you use only 0 and 180? Or only 90 and 270?
* While the paper is easy to understand, at times the writing is poor and awkward (e.g., opening sentence of intro, first sentence in section 2.2).

---

> ### Author Response · Authors · 2017-12-24
> **Response to AnonReviewer3 (part 1)**
>
> We thank the reviewer for the valuable feedback.  This is the 1st part of the answer to his/her comments.
>
> Comment:
> "Nearly all of the image categorization results (with the exception of those in Table 4) are presented for the contrived scenario where the unsupervised representation is learned from the same training set as the one used for the final supervised training of the categorization model. This is a useless application scenario. If labels for the training examples are available, why not using them for feature learning given that this leads to improved performance (see results in Tables)?"
>
> Answer:
> We disagree with the above statement that we do not present enough evaluation results for cases where a different training set is used between the unsupervised and supervised tasks.
>
> In Table 4 (which is Table 7 in the revised version of the paper) we evaluate the unsupervised learned features on THREE DIFFERENT PASCAL tasks: image classification, object detection, and object segmentation (the segmentation results were added in the revised version of the paper). These correspond to core problems in computer vision and evaluating the transferability of learned ConvNet features on those tasks is one of the most widely used and well-established benchmark for unsupervised representation learning methods [1,2,3,4].
>
> Moreover, in Figure 5.b we evaluate our unsupervised representation learning method on a semi-supervised setting when only a small part of the available training data are labelled and we demonstrate that our method can leverage the unlabelled training data to improve its accuracy on the test set.
>
> Furthermore, regarding the evaluation experiments that utilize the same training set for both the unsupervised and supervised learning (e.g. CIFAR-10 and ImageNet classification tasks), we note that this type of experiments have been proposed and are extensively used in all prior feature learning methods [1,2,3,4]. Therefore, they provide a well-established benchmark based on which we can compare to prior approaches. The reason why this is considered to be a useful benchmark is because it allows one to evaluate the quality of the unsupervised learned features by directly comparing them with the features learned in supervised way on the same training set (which provides an upper bound on the performance of the unsupervided features).
>
> ------
>
> Comment:
> "More importantly, this setup does not allow us to understand how general the unsupervised features are. Maybe they are effective precisely because they have been learned from images of the 10 classes that the final classifier needs to distinguish. I would have liked to see some results involving unsupervised learning from a dataset that may contain classes different from those of the final test classification or, even better, from a dataset of randomly selected images that lack categorical coherence (e.g., photos randomly picked from the Web, such as Flickr pics)."
>
> Answer:
> In general we believe that the primary goal of unsupervised representation learning is to learn image representations appropriate for understanding (e.g., recognizing or detecting) the visual concepts that were "seen" during training. Learning features that generalize on "unseen" visual concepts is indeed a desirable property but it is something that even supervised representation learning methods might struggle with it and is not the (main) scope of our paper. Nevertheless, as requested by the reviewer, we added in the revised version of our paper an evaluation of our unsupervised learned features on the scene classification task of Places205 benchmark (see Table 6). Note that for the scene classification results, the unsupervised features were learned on ImageNet that contains classes different from those of the scene classification task of Places205.
>
> ------
>
> [1] Richard Zhang et al, Colorful Image Colorization.
> [2] Jeff Donahue et al, Adversarial Feature Learning.
> [3] Noroozi and Favaro, Unsupervised learning of visual representations by solving jigsaw puzzles.
> [4] Piotr Bojanowski and Armand Joulin, Unsupervised Learning by Predicting Noise.

---

> ### Author Response · Authors · 2017-12-24
> **Response to AnonReviewer3 (part 2)**
>
> This is the 2nd part of the answer to the reviewer's comments.
>
> Comment:
> "In nearly all the experiments, the classifier is built on top of the frozen unsupervised features. This is in contrast with the common practice of finetuning the entire pretrained unsupervised net on the supervised task. It'd be good to know why the authors opted for the different setup and to see in any case some supervised finetuning results."
>
> Answer:
> We believe that this comment is inaccurate. First of all, for the PASCAL results in Table 4 (i.e., PASCAL classification, PASCAL detection, and the newly added PASCAL segmentation results), we FINETUNE THE ENTIRE NETWORK (see the last 3 columns of this table). Furthermore, in general for the experimental evaluation of our method on natural images (section 3.2), we want to emphasize that we follow THE SAME EVALUATION SETUP that prior unsupervised feature learning methods have used [1,2,3,4] and we did not propose a new one (this also allows us to compare with these approaches).
>
> Regarding the experiments presented in section 3.1 (that includes results in CIFAR-10), those were meant as a proof of concept of our work and to help better analyze various aspects of our approach before we move onto the more challenging, but also much more time consuming, experiments on ImageNet (see section 3.2). Therefore, for the experimental evaluation in CIFAR-10 we mimicked the evaluation setup that is employed by prior approaches on ImageNet (i.e., unsupervised feature learning on ImageNet and then training non-linear classifiers on top of them for the ImageNet classification task). Other than that we did not have any particular reason for not fine-tuning the learned features. However, as requested by the reviewer, we added  experiments with fine-tuning in the revised version of the paper (see Table 3). We observe that by fine-tuning the unsupervised learned features this further improves the classification performance, thus reducing even more the gap with the supervised case.
>
> ------
>
> Comment:
> "It would be useful to see the accuracy per class both when using unsupervised features as well as fully-supervised features. There are many objects that have a canonical pose/rotation in the world. Forcing the unsupervised features to distinguish rotations of such objects may affect the recognition accuracy for these classes. Thus, my request for seeing how the unsupervised learning affects class-specific accuracy."
>
> Answer:
> We added such results in Tables 8 and 9 (in appendix B) of the revised version of the paper.
>
> ------
>
> Comment:
> "While the results in Table 2 are impressive, it appears that the different unsupervised learning methods reported in this table are based on different architectures. This raises the question of whether performance gains are due to the better mechanism for unsupervised learning or rather the better network architecture."
>
> Answer:
> Indeed, each entry in Table 2 (Table 3 in the revised manuscript) has a different network architecture. It was not really possible for us to implement our method with each of those architectures and so those results are just indicative and not meant for direct comparison (we added this clarification in the revised manuscript - see caption of Table 3). The main bulk of experiments that directly compares our approach against other (more recent and more relevant) approaches is presented in section 3.2 of our paper. Regarding Table 2, we believe the most interesting and remarkable finding is the very small performance gap between our unsupervised feature learning method and the supervised case (that both use exactly the same network architecture).
>
> ------
>
> Comment:
> "I do understand that using only 0, 90, 180 and 270 degree rotations eliminates the issue of potentially recognizable artifacts. Nevertheless, it'd be interesting to see what happens empirically when the number of discrete rotations is increased, e.g., by including 45, 135, 225 and 315 degree rotations. And what happens if you use only 0 and 180? Or only 90 and 270?"
>
> Answer:
> We thank the reviewer for this suggestion. Please see Table 2 and relevant discussion in paragraph ``"Exploring the quality of the learned features w.r.t. the number of recognized rotations" (section 3.1) in the revised version of the paper.
>
> ------
>
> [1] Richard Zhang et al, Colorful Image Colorization.
> [2] Jeff Donahue et al, Adversarial Feature Learning.
> [3] Noroozi and Favaro, Unsupervised learning of visual representations by solving jigsaw puzzles.
> [4] Piotr Bojanowski and Armand Joulin, Unsupervised Learning by Predicting Noise.

---

> > ### Comment · AnonReviewer3 · 2018-01-10
> > **new experiments address most of my questions**
> >
> > I thank the authors for their response. The revised version of the paper includes several new experiments that address most of my questions. Specifically, I appreciate the following new analyses:
> > - performance vs number of rotations (Table 2);
> > - accuracy per class (Table 9);
> > - effect of fine-tuning (Table 3);
> > - generalization obtained by unsupervised learning on a dataset different from that used for subsequent supervised training (Table 6).
> >
> > One of my questions was about how much of the good performance of the method was due to learning the unsupervised features on the same training set used by the supervised learning. The newly-added Table 6 partly addresses this question. However, I would suggest to add to this table a baseline corresponding to training the RotNet unsupervised features on Places in order to have a direct comparison with the same features trained on ImageNet (last row).
> >
> > I am somewhat disappointed in the responses provided by the authors about two of the criticisms that I had raised: 1) doing unsupervised training on a training set involving only classes in the test set is a contrived setup, 2) lack of finetuning results. The authors respond to the former point by saying that my statement is inaccurate and to the latter by stating that they disagree with my point. I find both of these answers harsh and unnecessary. In my review I preambled both of my points by saying "... *nearly* all of the results ....". In fact, I did recognize in my review that Table 4 provides an exception to both of these points. My criticism was aimed at convincing the authors to run additional experiments around these two aspects. The revision contains new experiments that in my view significantly strengthen this work. Pointing to these new results would have sufficed....

---

### Official Review · AnonReviewer2 · 2017-12-15
**intuitive but effective self-supervised method (with some lack of evaluation thoroughness)**

**Rating:** 6
**Confidence:** 3

**Review:**

**Paper Summary**
    This paper proposes a self-supervised method, RotNet, to learn effective image feature from images by predicting the rotation, discretized into 4 rotations of 0, 90, 180, and 270 degrees. The authors claim that this task is intuitive because a model must learn to recognize and detect relevant parts of an image (object orientation, object class) in order to determine how much an image has been rotated.
They visualize attention maps from the first few conv layers and claim that the attend to parts of the image like faces or eyes or mouths. They also visualize filters from the first convolutional layer and show that these learned filters are more diverse than those from training the same model in a supervised manner.
	They train RotNet to learn features of CIFAR-10 and then train, in a supervised manner, additional layers that use RotNet feature maps to perform object classification. They achieve 91.16% accuracy, outperforming other unsupervised feature learning methods. They also show that in a semi-supervised setting where only a small number of images of each category is available at training time, their method outperforms a supervised method.
	They next train RotNet on ImageNet and use the learned features for image classification on ImageNet and PASCAL VOC 2007 as well as object detection on PASCAL VOC 2007. They achieve an ImageNet and PASCAL classification score as well as an object detection score higher than other baseline methods.
    This task requires the ability to understand the types, the locations, and the poses of the objects presented in images and therefore provides a powerful surrogate supervision signal for representation learning. To demonstrate the effectiveness of the proposed method, the authors evaluate it under a variety of tasks with different settings.



**Paper Strengths**
- The motivation of this work is well-written.
- The proposed self-supervised task is simple and intuitive. This simple idea of using image rotation to learn features, easy to implement image rotations without any artifacts
- Requiring no scale and aspect ratio image transformations, the proposed self-supervised task does not introduce any low-level visual artifacts that will lead the CNN to learn trivial features with no practical value for the visual perception tasks.
- Training the proposed model requires the same computational cost as supervised learning which is much faster than training image reconstruction based representation learning frameworks.
- The experiments show that this representation learning task can improve the performance when only a small amount of annotated examples is available  (the semi-supervised settings).
- The implementation details are included, including the way of implementing image rotations, different network architectures evaluated on different datasets, optimizers, learning rates with weight decayed, batch sizes, numbers of training epochs, etc.
- Outperforms all baselines and achieves performance close to, but still below, fully supervised methods
- Plots rotation prediction accuracy and object recognition accuracy over time and shows that they are correlated



**Paper Weaknesses**
- The proposed method considers a set of different geometric transformations as discrete and independent classes and formulates the task as a classification task. However, the inherent relationships among geometric transformations are ignored. For example, rotating an image 90 degrees and rotating an image 180 degrees should be closer compared to rotating an image 90 degrees and rotating an image 270 degrees.
- The evaluation of low-level perception vision task is missing. In particular, evaluating the learned representations on the task of image semantic segmentation is essential in my opinion. Since we are interested in assigning the label of an object class to each pixel in the image for the task, the ability to encode semantic image feature by learning from performing the self-supervised task can be demonstrated.
- The figure presenting the visualization of the first layer filters is not clear to understand nor representative of any finding.
- ImageNet Top-1 classification results produced by Split-Brain (Zhang et al., 2016b) and Counting (Noroozi et al., 2017) are missing which are shown to be effective in the paper [Representation Learning by Learning to Count](https://arxiv.org/abs/1708.06734).
- An in-depth analysis of the correlation between the rotation prediction accuracy and the object recognition accuracy is missing. Showing both the accuracies are improved over time is not informative.
- Not fully convinced on the intuition, some objects may not have a clear direction of what should be “up” or “down” (symmetric objects like balls), in Figure 2, rotated image X^3 could plausibly be believed as 0 rotation as well, do the failure cases of rotation relate to misclassified images?
- “remarkably good performance”, “extremely good performance” – vague language choices (abstract, conclusion)
- Per class breakdown on CIFAR 10 and/or PASCAL would help understand what exactly is being learned
- In Figure 3, it would be better to show attention maps on rotated images as well as activations from other unsupervised learning methods. With this figure, it is hard to tell whether the proposed model effectively focuses on high level objects.
- In Figure 4, patterns of the convolutional filters are not clear. It would be better to make the figures clear by using grayscale images and adjusting contrast.
- In Equation 2, the objective should be maximizing the sum of losses or minimizing the negative. Also, in Equation 3, the summation should be computed over y = 1 ~ K, not i = 1 ~ N.



**Preliminary Evaluation**
This paper proposes a self-supervised task which allows a CNN to learn meaningful visual representations without requiring supervision signal. In particular, it proposes to train a CNN to recognize the rotation applied to an image, which requires the understanding the types, the locations, and the poses of the objects presented in images. The experiments demonstrate that the learned representations are meaningful and transferable to other vision tasks including object recognition and object detection. Strong quantitative results outperforming unsupervised representation learning methods, but lacking qualitative results to confirm/interpret the effectiveness of the proposed method.

---

> ### Author Response · Authors · 2017-12-24
> **Response to AnonReviewer2 (part 1)**
>
> We thank the reviewer for the valuable feedback.  This is the 1st part of the answer to his/her comments.
>
> Comment:
> "The proposed method considers a set of different geometric transformations as discrete and independent classes and formulates the task as a classification task. However, the inherent relationships among geometric transformations are ignored. For example, rotating an image 90 degrees and rotating an image 180 degrees should be closer compared to rotating an image 90 degrees and rotating an image 270 degrees."
>
> Answer:
> We thank the reviewer for this suggestion on how to further improve the performance of our method (we will certainly try to explore if a modification of this type can be of any help). However, based on our intuition, we feel that the above modification to the rotation prediction task will most probably not have any positive effect with respect to the paper's goal of unsupervised representation learning simply because the rotation prediction task that we propose is used here just as a proxy for learning semantic representations. Moreover, it is debatable if the representations of two images that differ, e.g., by 90 degrees should always be closer than the representations of two images that differ by 180 degrees (if this is what the reviewer means). In any case, we would be glad to explore the above enhancement to our method proposed by the reviewer and report any positive findings in the final version of the paper.
>
> A first experiment that we did towards that direction is to modify the target distributions used in the cross entropy loss during the training of the rotation prediction task: more specifically, instead of using target distributions that place the entire probability mass on the ground truth rotation (as before), we used distributions that also allow some small probability mass to be placed on the rotations that differ only by 90 degrees from the ground truth rotation.
> However, this modification did not offer any performance improvement when the learned features were tested on the CIFAR-10 classification task. On the contrary, the classification accuracy was slightly reduced from 89.06 to 88.91.
>
> ------
>
> Comment:
> "The evaluation of low-level perception vision task is missing. In particular, evaluating the learned representations on the task of image semantic segmentation is essential in my opinion."
>
> Answer:
> We agree with the reviewer. Unfortunately we did not have this experiment ready before the submission deadline. However, we added now the segmentation results in the revised version of the paper (see Table 7); we observe that again our method demonstrates state-of-the-art performance among the unsupervised approaches.
>
> ------
>
> Comment:
> "In Figure 4, patterns of the convolutional filters are not clear. It would be better to make the figures clear by using grayscale images and adjusting contrast."
>
> Answer:
> In the revised version of the paper, we tried to improve the clarity of Figure 4 by further increasing the contrast.
>
> ------
>
> Comment:
> "The figure (Figure 4) presenting the visualization of the first layer filters is not clear to understand nor representative of any finding."
>
> Answer:
> However, we disagree with the statement that the visualizations of the 1st layer filters is not representative of any finding. It is true that the visualization of the 1st layer filters does not (directly) reveal the nature of the higher level features that a network learns, which is also what we are interested to understand. However, it very clearly demonstrates the nature of the low-level features that a network learns, which is also of interest, and, in our case, it shows that these features are very similar to those that a supervised object classification network learns. Due to the above reason, this type of visualization has been extensively used both for supervised methods [1] and for unsupervised methods [2,3,4].
>
> Furthermore, concerning the interpretation of the higher level features learned by our method, since it is difficult to provide a similar visualization as the one for the 1st layer filters, we choose to visualize instead the attention maps that those layers generate.
>
> ------
>
> [1] Alex Krizhevsky et al, ImageNet Classification with Deep Convolutional Neural Networks.
> [2] Jeff Donahue et al, Adversarial Feature Learning.
> [3] Noroozi and Favaro, Unsupervised learning of visual representations by solving jigsaw puzzles.
> [4] Piotr Bojanowski and Armand Joulin, Unsupervised Learning by Predicting Noise.

---

> ### Author Response · Authors · 2017-12-24
> **Response to AnonReviewer2 (part 2)**
>
> This is the 2nd part of the answer to the reviewer's comments.
>
> Comment:
> "ImageNet Top-1 classification results produced by Split-Brain (Zhang et al., 2016b) and Counting (Noroozi et al., 2017) are missing which are shown to be effective in the paper."
>
> Answer:
> The ImageNet Top-1 classification results for the NON-LINEAR classifiers of the Split-Brain and the Counting methods are missing because those methods do not report those results in their papers. However, in Table 4 (it is Table 7 in the revised version of the paper) we compare against the Split-Brain and Counting methods on the PASCAL tasks (i.e., classification, detection, and segmentation) and our method demonstrates state-of-the-art results. Furthermore, in the revised version of our paper we added the ImageNet Top-1 classification results for LINEAR classifiers of our method as well as prior methods that have reported such results (including Split-Brain and Counting methods) and again our approach achieves state-of-the-art results (see Table 5).
>
> -------
>
> Comment:
> "An in-depth analysis of the correlation between the rotation prediction accuracy and the object recognition accuracy is missing. Showing both the accuracies are improved over time is not informative."
>
> Answer:
> First we would like to clarify how we created the object recognition accuracy curve in Figure 5a and in general what Figure 5b demonstrates. In order to create the object recognition accuracy curve, in each training snapshot of RotNet (i.e., every 20 epochs), we pause its training procedure and we train from scratch (until convergence) a non-linear object classifier on top of the so far learned RotNet features (specifically the 2nd conv. block features). The object recognition accuracy curve depicts the accuracy of those non-linear object classifiers after the end of their training while the rotation prediction accuracy curve depicts the accuracy of the RotNet at those snapshots. Therefore, Figure 5a demonstrates the following fact: as the ability of the RotNet features for solving the rotation prediction task improves (i.e., as the rotation prediction accuracy increases), their ability to solve the object recognition task improves as well (i.e., the object recognition accuracy also increases).
>
> We also did another experiment towards clarifying the possible existence of a relation between the two tasks but this time we explored their relation in the opposite direction, i.e., we used input features learnt on the object recognition task in order to see their effectiveness on training a small network for the rotation prediction task.  Specifically, the rotation prediction network that we train on the CIFAR10 dataset has the same architecture as the 3rd (and last) conv. block of a NIN based ConvNet and this network is applied on top of the feature maps generated by the 2nd conv. block of a NIN based ConvNet trained on the object prediction task of CIFAR10. The rotation classification accuracy that this hybrid model achieves is 88.05, which is relatively close to the 93.0 classification accuracy achieved by a NIN based ConvNet trained solely on the rotation prediction task (and despite the fact that the first 2 conv. blocks of the hybrid model have been trained only with images of 0 degrees orientation).
>
> If the reviewer would like to specify any additional concrete experiment (reasonably easy to implement) that could be used to further clarify the existence of such a relation between the two tasks, we would be happy to implement and test it.
>
> ------

---

> ### Author Response · Authors · 2017-12-24
> **Response to Reviewer 2 (part 3)**
>
> This is the 3rd part of the answer to the reviewer's comments.
>
> Comment:
> "Not fully convinced on the intuition, some objects may not have a clear direction of what should be “up” or “down” (symmetric objects like balls), in Figure 2, rotated image X^3 could plausibly be believed as 0 rotation as well, do the failure cases of rotation relate to misclassified images?"
>
> Answer:
> Regarding the fact that some images might have ambiguous orientation, we note that this type of training examples comprise only a small part of the dataset and can essentially be seen as a small amount of label noise, which thus poses no problem for learning. On the contrary, the great majority of the used images have an unambiguous orientation. Therefore, the ConvNet, by trying to solve the rotation prediction task, will eventually be forced to learn object-specific features. This is also evidenced by the very strong experimental performance of these features when applied on a variety of different tasks including those of object recognition, object detection, object segmentation, and scene classification tasks (section 3 of the paper).
>
> Concerning the question posed by the reviewer if there is any connection between failure cases for rotation prediction and misclassifications w.r.t. object recognition, we did the following test in order to explore if there is any such correlation: first, we define as y0 a binary variable that indicates if an image is misclassified in the object recognition task by a fully supervised model, as y1 a binary variable that indicates if an image is misclassified in the object recognition task by our unsupervised learned RotNet model (by training a non-linear classifier on top of the RotNet features), and as x a continuous variable that indicates the fraction of rotations (out of the 4 possible ones per image) that are misclassified by RotNet. The point biserial correlation coefficient (https://docs.scipy.org/doc/scipy-0.14.0/reference/generated/scipy.stats.pointbiserialr.html) between the y1 and x variables on CIFAR-10 is 0.1473 with p-value 1.286e-49 while between the y0 and x variables is 0.1799 with p-value=1.5404e-73. Therefore, it seems that there is little correlation between failing to classify the rotations of an image and failing to classify the object that it depicts. Moreover, this holds regardless if we use a fully supervised object classifier (0.1799 correlation) or if we use an object classifier based on features learnt on the rotation prediction task.
>
> ------
>
> Comment:
> " “remarkably good performance”, “extremely good performance” – vague language choices (abstract, conclusion) "
>
> Answer:
> We rephrased the corresponding text to make it even more clear that the above statements relate to the state-of-the-art experimental results achieved by our method, which surpass prior approaches by a significant margin.
>
> ------
>
> Comment:
> "Per class breakdown on CIFAR 10 and/or PASCAL would help understand what exactly is being learned"
>
> Answer:
> We added such results in Tables 8 and 9 (in appendix B) of the revised version of the paper.
>
> ------
>
> Comment:
> "In Figure 3, it would be better to show attention maps on rotated images as well as activations from other unsupervised learning methods. With this figure, it is hard to tell whether the proposed model effectively focuses on high level objects."
>
> Answer:
> In the revised version of the paper in Figure 3 we added attention maps generated by a supervised model. By comparing them with those of our unsupervised model we observe that both of them focus on similar areas of the image in order to accomplish their task. Also, in Figure 6 (in appendix A), we added the attention maps of the rotated versions of the images. We observe that the attention maps of all the rotated images is roughly the same which means the attention maps are equivariant w.r.t. image rotations. This practically means that in order to accomplish the rotation prediction task the network focuses on the same object parts regardless of the image rotation.
>
> ------
>
> Comment:
> "In Equation 2, the objective should be maximizing the sum of losses or minimizing the negative. Also, in Equation 3, the summation should be computed over y = 1 ~ K, not i = 1 ~ N."
>
> Answer:
> We thank the reviewer for identifying the above typos. We fixed them in the revised version of the paper (see equation 3).
>
> ------

---

### Decision · Program_Chairs · 2018-01-29
**ICLR 2018 Conference Acceptance Decision**

**Decision:**

Accept (Poster)

**Comment:**

The paper proposes a new way of learning image representations from unlabeled data by predicting the image rotations. The problem formulation implicitly encourages the learned representation to be informative about the (foreground) object and its rotation. The idea is simple, but it turns out to be very effective. The authors demonstrate strong performance in multiple transfer learning scenarios, such as  ImageNet classification, PASCAL classification, PASCAL segmentation, and CIFAR-10 classification.